# Phthalate Esters Metabolic Strain *Gordonia* sp. GZ-YC7, a Potential Soil Degrader for High Concentration Di-(2-ethylhexyl) Phthalate

**DOI:** 10.3390/microorganisms10030641

**Published:** 2022-03-17

**Authors:** Tong Hu, Chen Yang, Zhengyu Hou, Tengfei Liu, Xiaotong Mei, Lianbao Zheng, Weihong Zhong

**Affiliations:** College of Biotechnology and Bioengineering, Zhejiang University of Technology, Hangzhou 310032, China; hutong1992@zjut.edu.cn (T.H.); 2111905019@zjut.edu.cn (C.Y.); 2112005170@zjut.edu.cn (Z.H.); dfliu2022@163.com (T.L.); 2111905025@zjut.edu.cn (X.M.); lianbao20211220@163.com (L.Z.)

**Keywords:** PAEs, comparative genomics analysis, DEHP, bioremediation

## Abstract

As commonly used chemical plasticizers in plastic products, phthalate esters have become a serious ubiquitous environmental pollutant, such as in soil of plastic film mulch culture. Microbial degradation or transformation was regarded as a suitable strategy to solve the phthalate esters pollution. Thus, a new phthalate esters degrading strain *Gordonia* sp. GZ-YC7 was isolated in this study, which exhibited the highest di-(2-ethylhexyl) phthalate degradation efficiency under 1000 mg/L and the strongest tolerance to 4000 mg/L. The comparative genomic analysis results showed that there exist diverse esterases for various phthalate esters such as di-(2-ethylhexyl) phthalate and dibutyl phthalate in *Gordonia* sp. GZ-YC7. This genome characteristic possibly contributes to its broad substrate spectrum, high degrading efficiency, and high tolerance to phthalate esters. *Gordonia* sp. GZ-YC7 has potential for the bioremediation of phthalate esters in polluted soil environments.

## 1. Introduction

Global increasing of plastic production and utilization, with only 9% of various plastic wastes being recycled, is leading to the accumulation and persistence of plastic debris in the environment [1]. As chemical plasticizers are commonly used in plastic products to improve the flexibility and workability of various plastic products [2], phthalate esters (PAEs) are also enriched in many natural environments, such as wastewater treatment plants [3], river water [4], and Chinese plastic film mulch soil [5,6,7,8]. Plastic debris can disperse throughout the oceans and their presence and distribution even exist in inhabited and remote locations of the world [9]. Thus, plastic debris has been recognized as a global health threat as the marine debris could be harmful to marine organisms and seabirds, and microplastics are detrimental to Chlorophyta and Ochrophyta most during growth or photosynthesis [10,11]. Also, PAEs had already been recognized as endocrine-disrupting chemicals (EDCs) which could interfere with the physiological endocrine function of animals and humans [12]. Nowadays studies have found that exposure to microplastics could enhance the antibiotic resistome of mice gut microbiota which would make traditional antibiotic treatments difficult and costly [13]. Thus, effective methods to eliminate PAEs pollution are essential. Until now, more than sixty kinds of PAEs have been produced and consumed in plastic production [14]. PAEs are differentiated by various side chains such as the simple-side-chain compound (diethyl phthalate, DEP and butyl benzyl phthalate, BBP), and complex-side-chain compound (di-(2-ethylhexyl) phthalate, DEHP) [14,15]. One of the most widely studied PAE compound is DEHP, which accounts for nearly 50% of the total PAE pollution production [16]. In addition, DEHP is found to have the highest concentration among other PAEs in water, air, soils and sediments.

The degradation strategies of PAEs include hydrolytic, photolytic, and microbial degradation [17]. As hydrolytic and photolytic methods work slowly and weakly, microbial degradation is regarded as the most promising strategy to solve PAE pollution, with many advantages such as high efficiency, fast kinetics, mild reaction conditions, and no secondary pollution [18]. The microbial degradation pathways of different PAEs were systematically identified. And it has been known that the PAEs’ degradation efficiency was influenced by pH, temperature, retention time, and microbial communities of the environment [19].

Two main steps are involved in the PAE biodegradation metabolic pathway according to the intermediates identified: (I) transformation of PAEs to phthalic acid (PA) and (II) complete degradation of PA [20].

Nowadays, more than 80 PAE-degrading strains isolated from various soil bodies or other environments have been identified, belonging to *Gordonia*, *Sphingomonas* [21], *Pseudomonas* [22], *Rhodococcus* [23] and *Comamonas* [24]. Some PAE-degrading strains can tolerate 500–2000 mg/L PAEs, but their application range is restricted due to their slow growth and reproduction, inability to tolerance high concentration PAEs, and poor environmental adaptability. Furthermore, PAEs in the natural environment usually exist in mixed states including different proportions of dibutyl phthalate (DBP), DEHP, DEP, BBP, di-n-octyl phthalate (DnOP), and dimethyl phthalate (DMP). DEHP, DEP, DBP, BBP, DnOP, and dimethyl phthalate (DMP) have been ranked as high-priority control pollutants by the US Environmental Protection Agency (USEPA) [25]. However, most reported PAE-degrading bacteria show limited substrates spectrum and degradation efficiency in soil or in aqueous environments. For example, *Gordonia polyisoprenivorans* G1 could degrade 243 mg/kg DEHP in soil to 50% over 2 days, while *Pseudomonas* sp. YJB6 could completely reduce 200 mg/L DBP in 3 days [22,26]. Therefore, isolating more PAE-degrading strains with stronger ability and higher efficiency to degrade a broader variety of PAEs is essential to directly apply or construct engineering chassis cells for PAEs bioremediation.

This study aimed to (1) isolate novel PAE-degrading strains exhibiting both higher efficiency and broader spectrum than reported strains, from Chinese PAE-polluted soil samples; (2) reveal the potential mechanism and degradation pathways of the PAE metabolism of isolated strains, based on comparative genomic analysis; (3) evaluate application potential of isolated strains for aqueous and soil environment bioremediation and for the construction of PAE-degrading chassis cell factories.

## 2. Materials and Methods

### 2.1. Chemicals and Media

Diethyl phthalate (DEP), dipropyl phthalate (DPrP), dibutyl phthalate (DBP), benzyl butyl phthalate (BBP), di-(2-ethylhexyl) phthalate (DEHP), dioctyl phthalate (DnOP), and diisononyl phthalate (DiNP) with 99% purity were purchased from Aladdin Chemistry Co., Ltd. (Shanghai, China). Methanol was HPLC grade while all other chemical reagents used in this experiment were analytical grade.

Luria-Bertani (LB) medium consisted of (g/L): NaCl 10, tryptone 10, yeast extract 5.

Basic inorganic salt medium (BSM) consisted of (g/L): K_2_HPO_4_·3H_2_O 1.0, NaCl 1.0, (NH_4_)_2_SO_4_ 0.5, MgSO_4_·7H_2_O 0.4, CaCl_2_ 0.0755, FeCl_3_ 0.0143.

For the preparation of solid media, 20 g/L agar was added to LB and BSM liquid mediums.

### 2.2. Isolation and Identification of Phthalate Esters Degrading Bacteria

Landfills are possible aggregation sites of PAEs pollutants from different plastic packaging waste and thus may be the most likely sources of PAE-degrading strains. Thus, we collected soil sample from the LaoHeiShan landfill, Liupanshui, Guizhou Province, China (39°52′48″ N, 105°30′5″ E), from which a 25 g soil sample was added into 20 mL sterile water for mixture completely with glass beads. Then 2 mL supernatant was added to 100 mL BSM medium containing 200 mg/L DEHP and the mixture was cultured under 30 °C and 180 rpm for 4 days. Then 2 mL mixture was transferred to 100 mL fresh BSM medium containing 200 mg/L DEHP for another round of 4 days culture. After several rounds, the cultures were gradient-diluted and spread onto the solid LB medium, and then incubated at 30 °C for 4 days. The single colonies were selected and inoculated into BSM medium containing different 200 mg/L PAEs, respectively, to test the PAEs’ degradation abilities. 

The morphology characteristics of pure colony were identified, while their 16S rRNA gene were sequenced (Tsingke Biotechnology Co., Ltd., Beijing, China) after amplified using universal primers 27F (5′-AGAGTTTGATCMTGGCTCAG-3′) and 1492R (5′-GGTTACCTTGTTACGACTT-3′). The 16S rRNA sequences comparison between the obtained strains and the reported strains in GenBank was conducted by BLAST and phylogenetic tree was then constructed using the neighbor-joining method with MEGAX software.

### 2.3. Culture Conditions and Analytic Method for Phthalate Esters

From slant, PAE-degrading bacterial strains were inoculated into LB medium and incubated under 30 °C and 180 rpm until the OD600 of broth reached 0.6–0.8. Cells were harvested by centrifugation at 6000 rpm for 3 min and washed with sterile water three times. Cells were then suspended for a final seed solution with the OD600 value of 0.8, which was inoculated into 50 mL BSM medium by a 2% (*v/v*) inoculation amount. Each group for control or treatment was in triplicate.

The broth was directly used to detect PAE content. Firstly, PAEs were extracted with an equal volume of dichloromethane, and then the organic phase was transferred to a rotary evaporator for drying. Then the samples were dissolved in 5 mL methanol and filtered through a 0.22 μm membrane filters. PAEs content was detected by HPLC (Agilent 1260) under the follow conditions: the UV wavelength of 235 nm; C18 column; the mobile phase contained 90% (vol) methanol and 10% (vol) water; the flow rate was 1.0 mL/min. 

### 2.4. Phthalate Esters Degradation Ability of Isolated Strains in BSM Medium

#### 2.4.1. Degrading Substrates Spectrum

To test the substrate spectrum of isolated PAE-degrading strains, five hundred milligrams of DEP, DPrP, DBP, BBP, DEHP, DnOP, and DiNP were added respectively in 1 L liquid BSM medium as the sole source of carbon and energy. After inoculation, all the cultures were incubated at 180 rpm and 30 °C for 2 days. Then the residue PAEs in the broth was detected by HPLC, while the cell growth OD600 was monitored by a spectrophotometer (UV5200, Shanghai Metash Instruments Co., Ltd., Shanghai, China). 

#### 2.4.2. Tolerance to High Concentrations of Phthalate Esters Degrading Strains

Five kinds of commonly used PAEs (DBP, BBP, DEHP, DnOP, and DiNP) were selected. The initial PAEs concentrations were set at 4 g/L. Strain was inoculated into liquid BSM medium containing DBP, BBP, DEHP, DnOP, and DiNP, respectively, and the residual concentrations of PAEs were determined on days 1, 3, and 5. 

#### 2.4.3. Mixed Phthalate Esters Degradation

PAEs usually exist in the natural environment as mixed format. Thus, four kinds of PAEs (200 mg/L DEP, DBP, DEHP, and DnOP6) were mixed to afford a final PEAs concentration of 800 mg/L in BSM medium. The cell growth (OD600) and residue PAEs concentration in cultures was measured at each 12-h intervals for 3 days. 

#### 2.4.4. Effects of Environmental Factors on Di-(2-ethylhexyl) Phthalate Degradation

A pH value range of 5, 6, 7, 8, 9, 10, a temperature range of 15 °C, 25 °C, 30 °C, 37 °C, 42 °C, and a NaCl content range of 2%, 4%, 6%, 8%, 10% (*v/v*), were selected to evaluate the effects of environmental factors on the DEHP degradation. The cell growth (OD600) and residue concentration of DEHP were detected after 2 days incubation at 180 rpm and 30 °C. The group without bacterial inoculation was set as control simultaneously.

#### 2.4.5. Degradation Kinetics of Di-(2-ethylhexyl) Phthalate

Under optimal conditions, the initial DEHP concentrations were set as 200, 500, 1000 and 2000 mg/L. The DEHP residue concentration in cultures was measured at each 12-h intervals for 3 days. In order to explore the effect of the initial concentration of DEHP on the degradation efficiency, the first-order kinetic equation was used to describe the degradation efficiency. The formula was as the following: lnC = −Kt + A, where the C is the DEHP concentration at time t. The K is the first-order constant, A is a constant. Therefore, the half-life of DEHP degradation could be expressed as: t1/2 = ln 2/K

### 2.5. Degradation of Di-(2-ethylhexyl) Phthalate in Soil

The soil was collected from a campus garden of the Zhejiang University of Technology (Hangzhou, China), in which large materials such as leaves, branches, roots, and stones, were removed. The soil sample was then sieved using a 2 mm mesh. The pH value and the water content of the soil were 7.19 and 17.79% (water to soil ratio at 2.5:1 *w/v*) respectively; thus, the soil sample was dried naturally at room temperature. Meanwhile, another sterilized part of the soil sample was prepared by autoclaving at 121 °C for 30 min. Then, 4 g/L DEHP acetone solution was added into natural soil and sterile soil respectively to a final concentration of 500 mg/kg. The resultant soil contaminated by DEHP was left overnight until the acetone had completely volatilized. Then, 40 g prepared soil sample was put in a 50 mL beaker and mixed with sterile water to restore the soil water content to the initial value (2.5:1 *w/v*). Two mL of the GZ-YC7 cell suspension, obtained in Section 2.3, were inoculated into 40 g of the abovementioned soil and then mixed completely. The mixture was then transferred into a 30℃ incubator. After 1, 2, 3, 4, and 5 days, two-gram soil samples were collected respectively, and stored at −80 °C. The freeze-dried soil samples were mixed with 3 mL methanol for 30 min ultrasonic dissolution, and soaked overnight to dissolve fully. The supernatant was collected by centrifugation at 12,000 rpm for 5 min and filtered via 0.22 μm membrane filter before HPLC detection for residue DEHP.

### 2.6. Genome Sequence Analysis of Strain GZ-YC7

The complete genome sequencing of GZ-YC7 was performed using a combined sequencing platform of DNBSEQ and PacBio at the Beijing Genomics Institute by BGI Genomics Co. Ltd., Shenzhen, China. Unavailable PacBio subreads (length < 1 kb) were removed. 

The Canu program was selected for self-correction. Then, draft genomic unitigs were assembled by Canu according to the high-quality corrected circular consensus sequence subreads set. And GATK (https://www.broadinstitute.org/gatk/; accessed on 14 March 2022) was taken to make corrections of single-base. Gene prediction of GZ-YC7 was performed using glimmer3 (http://www.cbcb.umd.edu/software/glimmer/; accessed on 14 March 2022) with Hidden Markov models. tRNA, rRNA and sRNAs were predicted according to tRNAscan-SE, RNAmmer, and the Rfam database. 

### 2.7. Gene Annotation and Protein Classification

The Blast alignment tool was chosen for function annotation. Seven databases such as KEGG (Kyoto Encyclopedia of Genes and Genomes), COG (Clusters of Orthologous Groups), NR (Non-Redundant Protein Database data bases), Swiss-Prot, and GO (Gene Ontology), TrEMBL and EggNOG were used for general function annotation. Virulence factors and resistance gene were identified based on the core dataset in VFDB (Virulence Factors of Pathogenic Bacteria), ARDB (Antibiotic Resistance Genes Database) database, and CAZy (Carbohydrate-Active Enzymes Database). All genome sequences used in this research were downloaded from NCBI database.

## 3. Results and Discussion

### 3.1. Isolation and Characterization of Strains

Among all isolated PAE-degrading strains, strain GZ-YC7 exhibited the broadest substrate spectrum and was then identified. Its colonies on LB solid plates appear orange color and its cell is round shape and Gram-positive (Figure 1a–c). GZ-YC7 could utilize glucose, fructose, xylose, mannose, and starch. GZ-YC7 showed positive results both on catalase and hydrogen sulfide experiments, but it showed negative results on urease, V-P, methyl red, and casein hydrolysis tests. 

The length of GZ-YC7 16S rRNA was 1417 bp and the sequence information has been uploaded to GenBank with the accession number of OM049462. It was found that GZ-YC7 shared 99% similarity to *Gordonia alkanivorans* strain 1960BRRJ (MK182084.1), and the corresponding phylogenetic tree was displayed in Figure 1c. Based on the phylogenetic analysis of 16S rRNA sequences, morphological, biochemical, and physiological characteristics, the strain GZ-YC7 was finally identified as a species belonging to Genus *Gordonia* and named as *Gordonia* sp. GZ-YC7. *Gordonia* sp. GZ-YC7 has been deposited in the China Center for Type Culture Collection (CCTCC), Wuhan University, China, with the number of CCTCC M2022045.

### 3.2. Phthalate Esters Degradation Ability of Isolated Strains in BSM Medium

#### 3.2.1. Degrading Substrate Spectrum

Most PAE-degrading strains always showed preference to finite substrates which might limit their applicability. For example, only DBP (1200 mg/L) was reported to be mineralized (approximately 90%) in 48 h by strain HD-1 which was enriched from activated sludge [27] Thus, the degrading substrate spectrum was an important evaluation criterion for the applied potential of PAE-degrading strains.

In this study, all seven kinds of PAEs could be efficiently degraded by *Gordonia* sp. GZ-YC7 within 48 h (Figure 2a). GZ-YC7 exhibited the best degrading efficiency on BBP and DEHP that were fully degraded, while its degradation ratio of DnOP and DiNP also reached 91.22% and 97.57%, respectively. However, the degradation ratio of short-chain PAEs (DBP, DPrP, and DEP) was only 88.02%, 66.77%, and 59.72%, respectively. 

It is generally assumed that PAEs containing long chains (DiNP, DnOP, DEHP) and benzene ring side chains (BBP) are hardly degraded because of steric hindrance [28]. For examples, the degradation rate of DBP was significantly higher than DEHP and DnOP by *Rhodococcus* sp. strain WJ4 [29]. *Arthrobacter* sp. ZJUTW can efficiently degrade DBP, while it cannot degrade DEHP [30]. However, some PAEs degrading strains showed versatile ability for PAEs with different side chains. For instances, *Rhodococcus* sp. 2G could degrade seven types of PAEs, but the degradation efficiency increased gradually with an increase in the side chain length [24]. *Gordonia alkanivorans* strain YC-RL2 was able to degrade all kinds of PAEs with short-chain length, long-chain length even with the benzene ring side chains; the degradation efficiency of DEHP with the concentration of 100 mg/L reached higher than 99% in 7 days [31]. 

*Gordonia* sp. GZ-YC7 also showed degrading ability of structurally diverse phthalate esters, especially long-side chains PAEs (DiNP, DnOP, and DEHP) and benzene ring side-chains (BBP), suggesting it has great potential for application in the future environmental bioremediation.

#### 3.2.2. Tolerance to High Concentrations of Phthalate Esters Degrading Strains 

The ability of GZ-YC7 to degrade and tolerate DBP, BBP, DEHP, DnOP and DiNP was measured under a high concentration of 4 g/L, which was the highest test concentration among the known test conditions in the previous publications (Table 1). The results (Figure 2b) showed that GZ-YC7 could degrade 70.71% of high concentration DEHP in 5 days, which was much higher than the DEHP degradation efficiency of *Gordonia alkanivorans* YC-RL2 (<10% of 4 g/L DEHP degraded in 7 days) [31]. As for DnOP and DiNP with long chains, the degradation efficiency of GZ-YC7 was 47.82% and 54.54%, respectively. In conclusion, GZ-YC7 could keep high degradation efficiency for high concentration PAEs, especially for DEHP, which indicated its higher tolerance than other strains and potential degradation to various PAEs.

#### 3.2.3. Mixed Phthalate Esters Degradation

PAEs usually present in coexisting forms in the natural environment; thus, the degradation ability of mixed PAEs is essential for a strain to be practically utilized. Four types of commonly used PAEs including DEP, DBP, DEHP, and DnOP were mixed equally for the degradation experiment (Figure 3a). It was found that DEP and DBP with short chains were degraded first while the longest-chain DnOP was degraded after 24 h cultivation. In addition, GZ-YC7 could degrade DEHP and DBP with the efficiency over 95% at 72 h, while the degradation efficiency of DEP and DnOP was 77.56% and 79.97%, respectively. Moreover, the degradation rate of DnOP in the mixed PAEs was significantly lower than that of only DnOP, suggesting the substrate preference of GZ-YC7 in PAEs mixed environment. Thus, the result illustrated that the PAEs degradation ability might be different in various environments even of the same strain, which could provide new insights for polluted environmental remediation.

#### 3.2.4. Effects of Environmental Factors on Di-(2-ethylhexyl) Phthalate Degradation

The effects of initial pH (Figure 2c), temperature (Figure 2d), and NaCl content (Figure 2e) on growth and DEHP degradation efficiency of GZ-YC7 were investigated in this study. Figure 2c showed that GZ-YC7 could degrade DEHP in the pH range of 5 to 10, and 500 mg/L DEHP was completely removed after 48 h incubation under pH 7.0 to 10.0. Meanwhile, the optimum pH for GZ-YC7 growth was eight. DEHP could be completely degraded by GZ-YC7 at temperature of 25 °C, 30 °C, and 37 °C, meanwhile, the optimum temperature for GZ-YC7 growth was 30 °C (Figure 2d). The concentration of NaCl showed a significant effect on DEHP degradation. GZ-YC7 could completely degrade DEHP at the concentration of 2% and 4% while the strain growth and DEHP degradation ability were significantly inhibited when the NaCl concentration increased above 6% (Figure 2e).

#### 3.2.5. Degradation Kinetics of Di-(2-ethylhexyl) Phthalate by Strain GZ-YC7

In some previous publications, it was found that PAEs degradation was inhibited by the increasing initial concentration of PAEs [39,40]. The initial DEHP concentration also influenced the growth of a DEHP-degrading *Pseudoxanthomonas* sp. N4. However, the strain N4 growth was obviously limited when the initial DEHP concentration decreased below 500 mg/L, while its growth stopped when the initial DEHP concentration increased above 500 mg/L [37]. 

Thus, the degradation ability of GZ-YC7 with the initial DEHP concentrations of 200, 500, 1000, and 2000 mg/L were also measured. The results (Figure 2f) showed that the degradation rate significantly decreased when the initial DEHP concentration increased from 200 to 2000 mg/L, meanwhile the half-life increased gradually (Appendix A). DEHP was completely degraded by GZ-YC7 within 24 h under the DEHP initial concentration of 500 mg/L, while GZ-YC7 growth remained until 36 h (Figure 2f). Only 87.11% DEHP was degraded in 3 days under the DEHP initial concentration of 2000 mg/L. However, GZ-YC7 exhibited a higher degradation efficiency and tolerant ability to high concentration of DEHP when compared to reported strains (Table 1). For examples, the DEHP degradation efficiency of GZ-YC7 was significantly higher than that of *Agromyces* sp. MT-O (35.0% of 1000 mg/L DEHP degraded within 7 days)*, Microbacterium* sp. CQ0110Y (half-life of 2000 mg/L DEHP was 2.36 day), and *Rhodococcus pyridinivorans* XB (87.5% of 800 mg/L DEHP degraded within 3 days) [26,32,33]. It was shown that high PAEs initial concentration could inhibit the DEHP degradation by GZ-YC7, but the half-life kept at a level from 0.45 to 1.49 days when the DEHP concentration increased from 200 mg/L to 2000 mg/L.

It was also found that most strains only grew and kept degrading in the concentration of DEHP lower than 1200 mg/L, but the DEHP degradation efficiency of *Gordonia* sp. GZ-YC7 was higher than any other DEHP-degrading strains (Table 1). When the DEHP concentration was below 500 mg/L, GZ-YC7 could completely remove the DEHP in one day while *Bacillus mojavensis* B1811 needed four days cultivation in 500 mg/L DEHP. When DEHP concentration reached 1000 mg/L, *Mycolicibacterium phocaicum* RL-HY01 had the highest DEHP degradation efficiency among the reported strains that DEHP was completely removed in 3 days, while GZ-YC7 only need 2.5 days. Overall, *Gordonia* sp. GZ-YC7 had the highest degradation efficiency under the DEHP concentration lower than 1000 mg/L. Most DEHP concentration in natural environments were at low level (<1000 mg/L), the new DEHP-degrading strain *Gordonia* sp. GZ-YC7 with the highest DEHP degradation efficiency indicated greater potential in environmental governance.

In addition, natural environments were diverse and dynamic, where sometimes the concentration of DEHP might be much higher than conventional condition. Thus, strains with high tolerance ability of DEHP concentration had wider applications and could provide new ideas for the remediation of extreme polluted environments. The DEHP concentration of 4000 mg/L degraded by *Gordonia* sp. GZ-YC7 was known to be the highest DEHP concentration. The results showed that GZ-YC7 could still grow in 4000 mg/L DEHP and DEHP degradation reached 70.71% after 5 days. It indicated that *Gordonia* sp. GZ-YC7 had extreme DEHP stress tolerance with the highest degradation efficiency under low DEHP concentration (<1000 mg/L) and the strongest tolerance in 4000 mg/L DEHP. Thus GZ-YC7 was a suitable choice for DEHP biodegradation. 

### 3.3. Degradation of di-(2-ethylhexyl) Phthalate in Soil

DEHP was more easily enriched in soil than in water and led to serious soil contamination [26]. Thus, the DEHP degradation efficiency of strain GZ-YC7 in soil is necessary to evaluate. The time course of DEHP concentration changes in soil samples is showed in Figure 3b. No significant change of DEHP concentration was detected in both group NSS and group SS within 5 days. After the inoculation of strain GZ-YC7, 45%, 22%, and 47.33% DEHP with the initial concentration of 500 mg/kg were degraded in groups NSS7 and SS7. It was found that there was no significant difference of DEHP degradation by GZ-YC7 with or without soil sterilization, suggesting that there was no synergy of GZ-YC7 with native microorganisms in the soil and only GZ-YC7 played the role of DEHP degrader in soil samples [33,41]. In previous publications, it was found that the soil microbial activity was inhibited when DEHP concentration reached 100 mg/kg [27,32,38,42]. However, GZ-YC7 could still grow and efficiently degrade 500 mg/kg DEHP in soil. GZ-YC7 has potential for application in high concentration DEHP polluted environments.

### 3.4. Genome Sequencing and Analysis

To date, the complete genomic information of only six *Gordonia* sp. strains including WA4-43, KTR9, JH63, 135, PDNC005 and YC-JH1, have been uploaded to NCBI (11 January 2022). The whole genome circle map of GZ-YC7 was completed by bioinformatics analysis after the quality control (Appendix A). The sequencing data statistics of DNBSEQ and PacBio are displayed in Appendix A. 

The complete genome of strain GZ-YC7 was identified to be 5,027,874 bp with the GC content of 67.93%, and there was no plasmid detected in GZ-YC7. This genome revealed 4654 predicted genes including 72 non-coding RNA (ncRNA) (Appendix A). A total of nine sRNA candidates were predicted, which may regulate some specific biological functions such as biofilm formation, iron metabolism and so on [43]. All genes were annotated against twelve databases including VFDB, ARDB, TREMBL, CAZY, IPR, Swiss-Prot, COG, CARD, GO, KEGG, NR, and T3SS. The details of all above genome data are displayed in Appendix A.

### 3.5. Comparison of Esterase Enzymes

In the process of catalysis of PAEs by esterase (Appendix A), esterase could be divided into three types. As shown in Table 2, seven putative esterase genes were predicted in GZ-YC7 genome based on reported esterase genes in previous publications (Appendix A). The gene GZ-YC7GL001189 of GZ-YC7 shared 98.94% similarity to the reported gene *mehpH* in *Gordonia* sp. P8219 [44]. The similarity between GZ-YC7GL004260 and the gene *estS1* of *Sulfobacillus acidophilus* DSM10332 was only 31.62% [2]. The phylogenetic tree of the seven putative esterase genes and other reported esterase genes was constructed as in Appendix A. It was confirmed that three types of esterase genes simultaneously exist in GZ-YC7 genome. The diversity of esterase enzymes existing in GZ-YC7 might play a role in the degradation ability of various PAE compounds.

Eight putative esterase enzymes were also predicted in *Gordonia alkanivorans* YC-RL2. However, only four of the eight enzymes showed more than 30% identity: (1) WP_006868835.1, 32% identity to Est1 of *Sulfobacillus acidophilus* DSM10332; (2) WP_005200181.1, 38% similarity with EstG of *Sphingobium* sp. SM42; (3) WP_006358366.1 and (4) WP_006358508.1, 35% and 37% identity to CarEW of *Bacillus* sp. K91, respectively [30]. 

### 3.6. Genetic Peculiarities and Pathway of Phthalate Esters Degradation by GZ-YC7

It has been reported that the *pht* cluster, *pca* cluster, *ben* cluster, and cat clusters participated in the process of PAEs biodegradation [24,31,49]. After PAEs were catalyzed to phthalic acid (PA) by esterase, the metabolism from PA to TCA cycle was catalyzed by gene clusters *pht*, *pca*, *ben*, and *cat* (Figure 4a). However, no *pht* gene cluster was detected in *Rhodococcus* sp. 2G which contains whole pca gene cluster, *ben* gene cluster, and *cat* gene cluster annotated in the genome. Similarly, no *pht* gene cluster was detected in *Gordonia alkanivorans* YC-RL2 which contains only *ben* gene cluster and *cat* gene cluster annotated in the genome. But what really distinguished *Gordonia* sp. GZ-YC7 is that all gene clusters of *pht*, *pca*, *ben*, and *cat* were detected in the genome. Two metabolism pathways of PAEs predicted previously are displayed in Figure 4b [24]. It can be seen that genes participating in these two pathways all exist in the GZ-YC7 genome, which suggests that two metabolism pathways might coexist in GZ-YC7. In a word, these two genetic peculiarities of GZ-YC7 may contribute to its high degradation efficiency of PAEs.

The gene distribution comparison of the above gene clusters between *Gordonia* sp. GZ-YC7 and other strains (Figure 5) showed that the gene distribution of *phtAa*, *phtAb*, *phtAc*, and *phtAd* belonging to *pht* cluster in *Gordonia* sp. GZ-YC7 were the same to that in *Gordonia* sp. HS-NH1 and *Gordonia* sp. YC-JH1. The *pca* cluster of *Gordonia* sp. GZ-YC7 shared the highest similarity with *Gordonia* sp. YC-JH1. However, more transport genes (*benM*, *benE*, *benK*) involved in ben cluster were annotated in GZ-YC7, while *benM* exists only within the cat gene cluster in GZ-YC7 genome. These transport genes could influence the benzoate metabolic process. All the above genetic peculiarities and coexisting various esters contribute to the substrate diversity and high degrading efficiency of GZ-YC7. Thus, GZ-YC7 could provide abundant raw materials for the construction of totipotent PAE-degrading strains. 

PAEs metabolites gene clusters analysis showed that *Gordonia* sp. GZ-YC7 possessed complete metabolic pathways to degrade various PAEs, which also contributes to the wide substrate spectrum of PAEs degradation. In order to have deeper insight into the genetic background of the new PAE-degrading strain *Gordonia* sp. GZ-YC7, all the related pathways were predicted and analyzed. According to the annotation of genes associated with PAEs biodegradation processes, the putative pathways were proposed as Figure 4b. Although genes encoding decarboxylases which could convert PA (Phthalate) to benzoate were not discovered in GZ-YC7, the pathway catalyzed by *pca* gene cluster was complete [35,49]. PA would be converted to phthalate 3,4-cis-dihydrodiol and finally to Acetyl-CoA, which then entered the TCA cycle.

## 4. Conclusions

*Gordonia* sp. GZ-YC7 exhibited the highest degradation efficiency among all reported strains under the DEHP concentration lower than 1000 mg/L. Furthermore, GZ-YC7 could maintain high PAE degradation efficiency even with a high PAE concentration of 4 g/L under optimal conditions (pH 7.0, 30 °C, and 6% NaCl), suggesting the highest tolerance to PAEs among all reported strains. The diversity of esterases and metabolic pathways predicted in the *Gordonia* sp. GZ-YC7 genome, also supports its great potential for application in the remediation of contaminated soil.

## Figures and Tables

**Figure 1 microorganisms-10-00641-f001:**
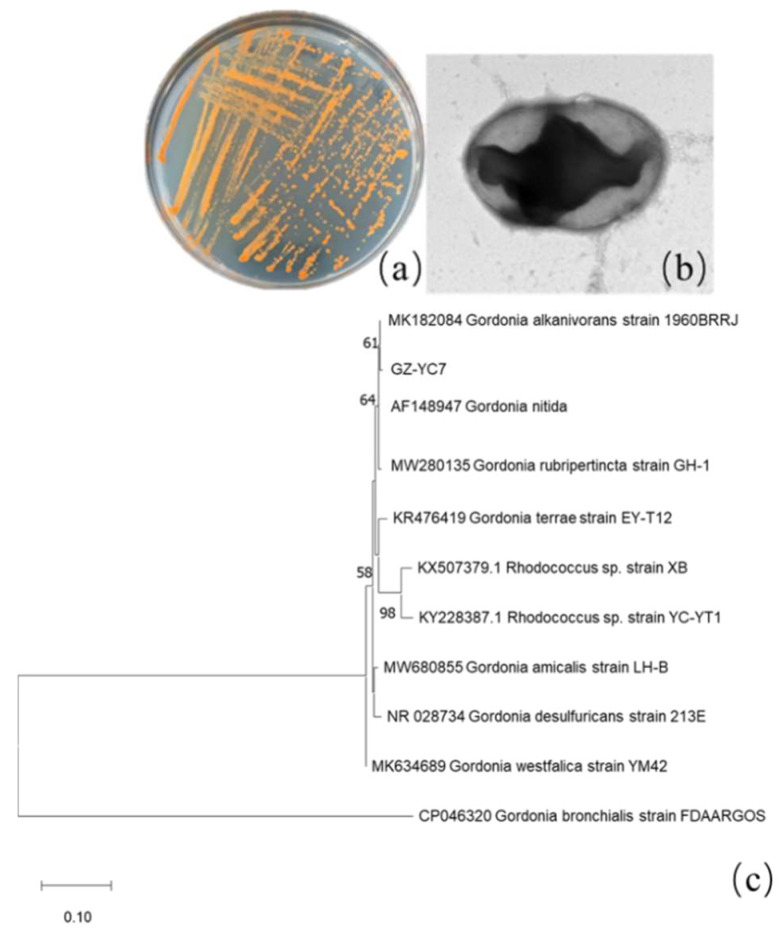
GZ-YC7 morphology of (**a**) colonies on the LB solid plate, (**b**) cell under TEM (60,000×), (**c**) phylogenetic tree analysis based on 16S rRNA sequences, Neighbor-Joining method with a bootstrap value of 1000.

**Figure 2 microorganisms-10-00641-f002:**
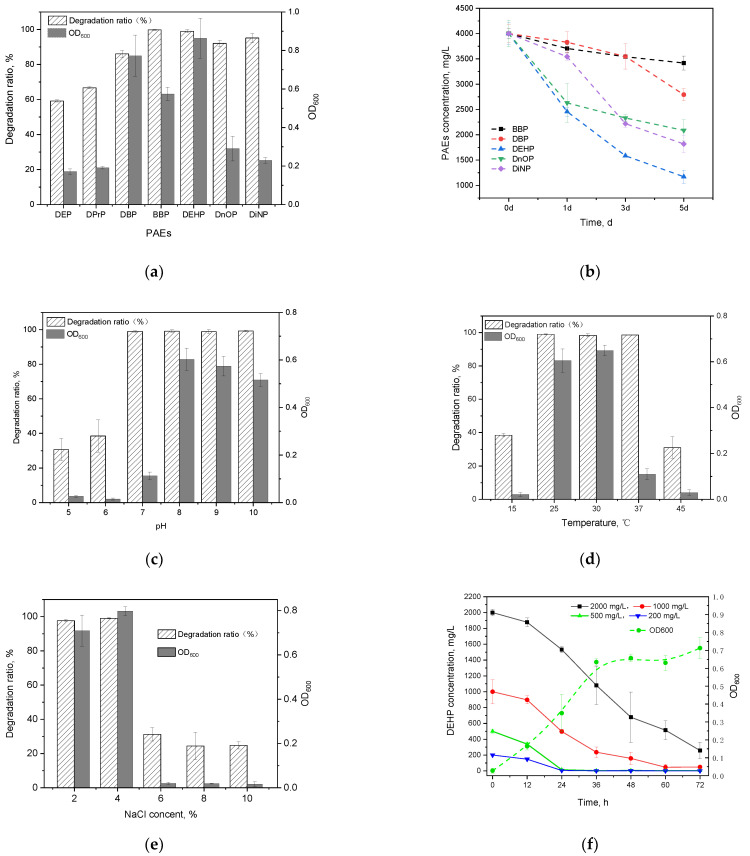
PAEs degradation and the growth of strain GZ-YC7 under 500 mg/L concentration of PAEs for 2 days (**a**); different PAEs degradation ability of GZ-YC7 under a high concentration of 4000 mg/L (**b**). DEHP degradation ratio and the growth of strain GZ-YC7 under different pH (**c**), temperature (**d**), NaCl concentration (**e**), and DEHP initial concentration with growth curves of GZ-YC7 under 500 mg/L DEHP (**f**).

**Figure 3 microorganisms-10-00641-f003:**
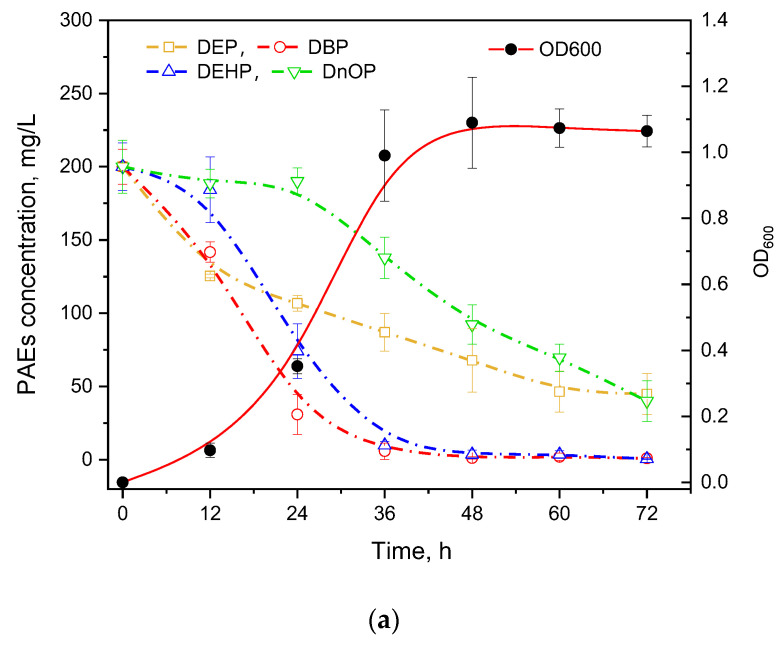
(**a**) PAEs Degradation and growth ability of GZ-YC7 in the BSM containing 800 mg/L mixed four PAEs (200 mg/L of DEP, DBP, DEHP, and DnOP); (**b**) DEHP degradation by GZ-YC7 in soil. NSS: Non-sterilized soil without GZ-YC7; NSS7: Non-sterilized soil with GZ-YC7; SS: Sterilized soil without GZ-YC7; SS7: Sterilized soil with GZ-YC7.

**Figure 4 microorganisms-10-00641-f004:**
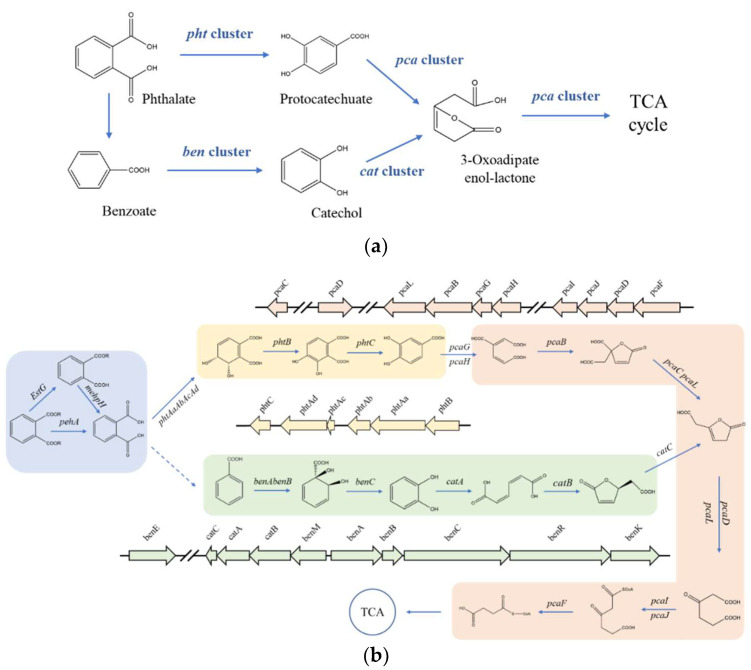
Gene clusters (**a**) the putative pathways (**b**) of PAEs degradation and in *Gordonia* sp. GZ-YC7. Module 1 in ** blue** is catalyzed by esters; Module 2 in **yellow** is catalyzed by the *pht* gene cluster; Module 3 in **green** is catalyzed by the ben and *cat* gene clusters; Module 4 in **red** is catalyzed by the *pca* gene cluster.

**Figure 5 microorganisms-10-00641-f005:**
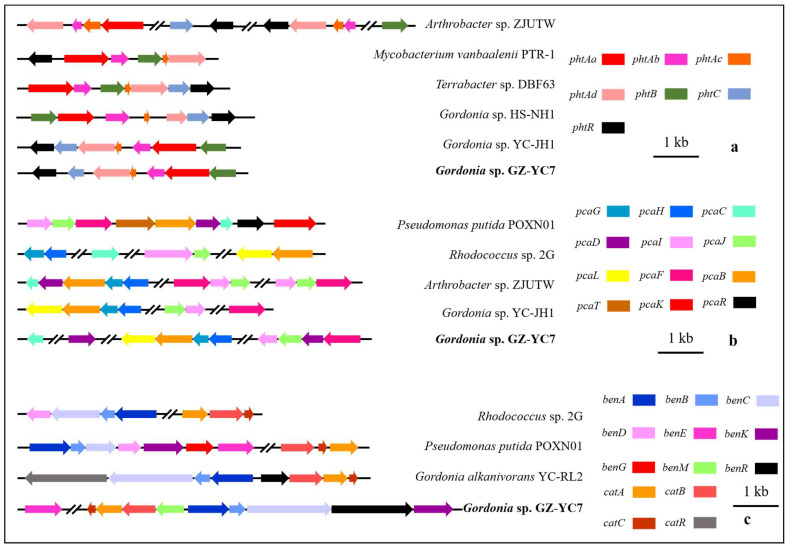
The comparative genomics analysis of the gene clusters *pht* (**a**), *pca* (**b**), *ben* (**c**), and *cat* (**c**).

**Table 1 microorganisms-10-00641-t001:** Comparison between the strain GZ-YC7 and reported DEHP-degrading strains.

Strain	Source	Degrading Substrates Spectrum	DEHP Degrading Efficiency	References
*Rhodococcus* sp. 2G	activated sludge	DEHP, DMP, DEP, DBP, BBP, DnOP, DiNP	200 mg/L, 5 day, >95%	[24]
*Rhodococcus ruber* YC-YT1	marine plastic debris	DEHP, DDP, DNP, DOP, DCHP, BBP, DHPP, DHP, DAP, DBP, DPrP, DEP, DMP	100 mg/L, 3 day, >95%	[12]
*Rhodococcus pyridinivorans* XB	activated sludge	DEHP, DMP, DEP, DBP	400 mg/L, 3 day, 100%	[32]
*Agromyces* sp. MT-O	landfill soil	DEHP, DMP, DEP, DBP, DnOP	1000 mg/L, 7 day, 65%	[33]
*Bacillus mojavensis* B1811	soil	DEHP, DEP, DMP, DBP, BBP, DnOP, DPP	500 mg/L, 4 day, 100%	[34]
*Pseudarthrobacter defluvii* E5	agricultural soil	DEHP, DMP, DEP, DBP, DHXP	1200 mg/L, 2 day, >50%	[35]
*Mycolicibacterium phocaicum* RL-HY01	wastewater	DEHP, DMP, DEP, DBP	1000 mg/L, 3 day, 100%	[36]
*Achromobacter* sp. RX	activated sludge	DEHP	300 mg/L, 4 day, 96%	[14]
*Pseudoxanthomonas* sp. N4	Denitrification biofilter reactor	DEHP	1250 mg/L, 5 day, 30%	[37]
*Gordonia* sp. Lff	river sludge	DEHP, DMP, DEP, DBP, DOP	2000 mg/L, 3 day, 91.4%	[38]
*Gordonia alkanivorans* YC-RL2	soil	DEHP, DCP, DEP, DMP, DBP	1000 mg/L, 7 day, 68.3%	[31]
*Gordonia* sp. GZ-YC7	landfill soil	DEHP, DEP, DPrP, DBP, BBP, DnOP, DiNP	**4000 mg/L, 5 day, 70.71%;** **2000 mg/L, 3 day, 87.11%;** **1000 mg/L, 2.5 day, 100%; 500 mg/L, 1 day, 100%;** **200 mg/L, 1d, 100%**	This study

^1^ Red represents PAEs with long side chains; green represents PAEs with short side chains; blue represents PAEs with cyclic side chains.

**Table 2 microorganisms-10-00641-t002:** Alignment results of esters in *Gordonia* sp. GZ-YC7.

Type	*Ester*	Strain	Gene Accession Number in GZ-YC7	Similarity (%)	References
**I**	*EstS1*	*Sulfobacillus acidophilus* DSM10332	GZ-YC7GL004260	31.62	[2]
**I**	*EstSP1*	*Sphingomonas glacialis* PAMC 26605	GZ-YC7GL000423	46.62	[45]
**II**	*mehpH*	*Gordonia* sp. P8219	GZ-YC7GL001189	98.94	[44]
**II**	*patE*	*Rhodococcus jostii* RHA1	GZ-YC7GL000190	38.92	[46]
**III**	*EstG*	*Sphingobium* sp. SM42	GZ-YC7GL000793	40.37	[47]
**III**	*CarEW*	*Bacillus* sp. K91	GZ-YC7GL000873	37.16	[48]
**III**	*pehA*	*Arthrobacter* sp. ZJUTW	GZ-YC7GL001562	34.07	[30]

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
