# Peer review of "Phthalate Esters Metabolic Strain Gordonia sp. GZ-YC7, a Potential Soil Degrader for High Concentration Di-(2-ethylhexyl) Phthalate"

_microorganisms, 2022, doi:10.3390/microorganisms10030641_

Round 1
Reviewer 1 Report
A report for: microorganisms-1646690-Phthalate esters metabolic strain Gordonia sp. GZ-YC7, a potential soil degrader for high concentration di-(2-ethylhexyl) phthalate
-It is an interesting and a really technical report. However, I find some flaws in the manuscript e.g. in the methodology and in the interpretation of the results.I am afraid it is difficult to recommend this manuscript for publication in the present form.
-Please rewrite the abstract.
-The justification for objectives selection is needed.
-Line 90 Soil sample was collected from the LaoHeiShan landfill, Liupanshui, Guizhou Prov- 91 ince, China (39° 52′ 48″ N, 105° 30′ 5″ E), from which 25 g soil sample was added into 20 92 mL sterile water for mixture completely with glass beads. Please provide more details of th soils.
-Please improve the figure 1.
-Section 4 Discussion is rather part of the conclusions. Please write a new conclusions section (adjust to the results obtained).
-Please follow the rules of the journal especially in the references section.
I hope you find my review useful.

Reviewer 2 Report
I do find this work interesting and valuable, however there are some issues which should be clarified.
Lines 34 - Shouldn’t be „exposure”?
Paragraph 2.5. – I didn’t find here the information about adding GZ-YC7 – see the data in Figure 3b
„Results” section should be rather named „Results and discussion”,as you dicussed here your results.
Line 236 – which previous publications?
Line 234 – there aren’t any data about DiNP in Figure 2b.
Table 1 – correct please, so the heading „DEHP degrading……”is visible
Line 348 – please correct way of referencing - into number
Line 430 – it should be rather named „Conclusions”
